# Introduction of a sEMG Sensor System for Autonomous Use by Inexperienced Users

**DOI:** 10.3390/s20247348

**Published:** 2020-12-21

**Authors:** Elisa Romero Avila, Elmar Junker, Catherine Disselhorst-Klug

**Affiliations:** Department of Rehabilitation & Prevention Engineering, Institute of Applied Medical Engineering, RWTH Aachen University, Pauwelsstr. 20, 52074 Aachen, Germany; romero@ame.rwth-aachen.de (E.R.A.); junker@ame.rwth-aachen.de (E.J.)

**Keywords:** surface electromyography, sensor system, dry electrodes, wearable device, monitoring, rehabilitation, dynamic contractions, activities of daily life, usability

## Abstract

Wearable devices play an increasing role in the rehabilitation of patients with movement disorders. Although information about muscular activation is highly interesting, no approach exists that allows reliable collection of this information when the sensor is applied autonomously by the patient. This paper aims to demonstrate the proof-of-principle of an innovative sEMG sensor system, which can be used intuitively by patients while detecting their muscular activation with sufficient accuracy. The sEMG sensor system utilizes a multichannel approach based on 16 sEMG leads arranged circularly around the limb. Its design enables a stable contact between the skin surface and the system’s dry electrodes, fulfills the SENIAM recommendations regarding the electrode size and inter-electrode distance and facilitates a high temporal resolution. The proof-of-principle was demonstrated by elbow flexion/extension movements of 10 subjects, proving that it has root mean square values and a signal-to-noise ratio comparable to commercial systems based on pre-gelled electrodes. Furthermore, it can be easily placed and removed by patients with reduced arm function and without detailed knowledge about the exact positioning of the sEMG electrodes. With its features, the demonstration of the sEMG sensor system’s proof-of-principle positions it as a wearable device that has the potential to monitor muscular activation in home and community settings.

## 1. Introduction

Wearable sensors have increasingly been incorporated and used in the rehabilitation of patients who suffer from diseases with associated movement restrictions. The advantage here is that wearable sensors provide real-world data on patients’ movement performance. Wearable sensors are used in various contexts, both for monitoring purposes at home and in community settings. Applications focus on the assessment of treatment efficacy, early detection of disorders, prevention and home rehabilitation [1,2] as well as the provision of an input variable for the control of prostheses and assistive systems such as robots and intelligent orthoses [2,3]. Quantitative measures enable the assessment of treatment efficacy and are an important tool with which clinicians can tailor therapy to the individual needs of their patients as they provide information about the patients’ performance between two therapeutic sessions [4]. This not only allows remote monitoring by the therapist, but can also give direct feedback to the patient [1].

Besides physiological and biochemical sensing, the detection of motion with wearable sensors is particularly prevalent. Here, accelerometers or inertial measurement units [5] are mainly used, which allow statements to be made about the activity, movement execution, and compensatory movement strategies. Reliably tracking changes in motor function yields useful information to guide the rehabilitation process of patients with disease related movement restrictions [6,7,8]. Do Nascimento et al., (2020) were able to show in their review paper that IMUs are among the most frequently used sensors in rehabilitation with a share of 14% [9]. The authors explain this high proportion by the increasing availability of IMU sensors, the decreasing cost of these devices, and their increasingly proven effectiveness in remote monitoring and supervision of exercises and everyday activities. Less frequently, in only 7% of all publications in the field of rehabilitation, were surface electromyography sensors (sEMG sensors) used, whereby no distinction is made by do Nascimento et al. between sEMG sensors using self-adhesive pre-gelled sEMG electrodes and dry electrodes [9]. However, particularly in the case of neurological diseases associated with motor impairments, information about the activation of muscles during the performance of tasks is crucial [10,11]. Furthermore, it has been shown that a combination of IMUs and sEMG recording allows not only the detection and classification of everyday movements but also a more reliable monitoring of the execution of therapeutic exercises [12,13]. Despite the potential gain in therapeutic information, sEMG has surprisingly not achieved the importance in rehabilitation that it could [14].

There are certainly many reasons why the translation of sEMG into clinical application is limited [14]. In the case of applications for (remote) monitoring of the execution of exercises and activities of daily living, user-friendliness and usability certainly play a central role [2,11,15,16]. Here, self-adhesive pre-gelled electrodes, which are usually used in surface electromyography, quickly reach to their applicability limits. Although self-adhesive pre-gelled electrodes provide good sEMG signal quality with a high signal-to-noise ratio and low motion artifacts even in highly dynamic movement conditions [17], their application usually requires the assistance of a trained operator. However, for monitoring the execution of therapeutic exercises or activities of daily living it would be desirable if the sEMG sensors can be attached by the patients without assistance, after a first introduction by the therapist (expert). For an autonomous application of sEMG by the patient, sEMG sensors using dry electrodes are more appropriate. Different types of dry electrodes are described in literature. They differ mainly in electrode shape and electrode material. Common to all of them is that the signal-to-noise ratio is worse and the exposure to interference is higher compared to pre-gelled electrodes. Additionally, the lower signal energy and the associated vulnerability to interference makes it necessary to amplify the sEMG signal close to the electrode [17]. In such active electrodes the dry electrode is mounted onto an operational amplifier forming an integrated sEMG sensor [18]. Such active sEMG electrodes are increasingly used in rehabilitation applications. Among many others, some examples can be found in [5,12,13,19,20,21,22,23].

In order to enable autonomous use of the sEMG sensor by the patient in rehabilitation settings, the sensor design has to meet specific requirements. These concern on the one hand patients’ ability to apply the sensor by themselves and on the other hand the minimization of the risk of a misplacement of the sensor. With regard to the first requirement, it should be noted that the functionality of the upper extremity may be considerably limited, especially in patients with movement restrictions due to neurological diseases. To ensure usability by patients with limited arm function, the sEMG sensor should be attachable with one hand. This is achieved with many of the active sEMG sensors currently used in rehabilitation [5,13,19,20,22,23,24]. The disadvantage of such ‘easy-to-apply’ sensors is that they can be placed anywhere on the limb. This increases the risk of incorrect placement of the electrodes relative to the muscle by the patient and affects the quality of the sEMG signal and thus the diagnostic value of the data collected, which depends largely on the correct placement of the electrodes relative to the muscle being examined [17,25,26,27,28,29,30]. An incorrect placement of the electrodes has significant effects on the sEMG signal amplitude, the frequency content of the signal and on the degree of crosstalk between adjacent muscles. Some sEMG sensors currently used in rehabilitation settings meet this challenge with electrodes that have a larger active surface or higher inter-electrode distances [29,31,32]. However, this implies a low pass filtering of the signal and, consequently a loss of significant information [33] as well as an increased risk of crosstalk [34]. In the SENIAM recommendations, clear specifications were made regarding electrode size and inter-electrode distance, which should be kept to ensure that the sEMG signal is correctly derived and thus remains interpretable [35]. According to SENIAM, the inter-electrode distance should not exceed 20 mm, which implies that the dimension of the electrode in the direction of the muscle fiber must be less than 10 mm [30,35]. The SENIAM recommendations, originally intended for isometric contractions, have been extended to dynamic conditions by the international CEDE project [30].

The second challenge sEMG faces when monitoring exercises and everyday movements is that the signal has to be recorded in non-isometric conditions. This not only creates special requirements for signal processing [36], it also requires that non-isometric conditions be taken into account when designing the sEMG sensor. sEMG signals recorded under dynamic conditions carry a high risk of motion artifacts, which are difficult to remove by filtering. These movement artifacts are caused either by movement between the excited muscle fibers, as the source of the sEMG signal, relative to the electrodes, movement of the electrodes relative to the skin surface, or by cable movement [17,18]. The latter can be reduced by using active electrodes. Since active electrodes have both amplifiers and electrodes in a rigid unit, the sEMG sensor has a higher composite weight compared to systems using self-adhesive pre-gelled electrodes. Due to the higher inertia of such a sEMG sensor and since the soft tissue between the source and the electrode is not rigid but can be displaced, the sEMG sensor does not directly follow, in direction and magnitude, changes in velocity of the human body. The result is a higher risk of motion artifacts, which heightens with increasing dynamics of the investigated movement. This risk can be counteracted by distributing the masses of the sEMG sensor in such a way that the moment of inertia of the entire sensor system is reduced. Such a mass distribution is found in sEMG armbands such as the Myo Armband by Thalmic Labs or the 3DC Armband from the University of Laval [21,37]. However, both these armbands are designed for gesture recognition and are used for prosthetic control or interaction in virtual environments [37]. The algorithms for gesture recognition are usually based on a pattern recognition process that requires less sEMG signal quality compared to applications in diagnostics and therapy. The signal quality of the Myo armband in particular is too poor to provide detailed and reliable information on muscular activation, due to its low sampling rate of 200 Hz and an electrode configuration that does not comply with the SENIAM recommendations [17,18,30,35].

To summarize, the autonomous use of sEMG sensors by the patient in the context of rehabilitation is still a problem. Three particular challenges are to ensure easy handling of the device, if possible with one hand, sufficiently correct positioning of the sensor relative to the muscle and to avoid motion artifacts. Presently, there is no sEMG sensor or sensor system available that meets the requirements for usability and signal quality. This paper aims to demonstrate the proof-of-principle of an innovative sEMG sensor system, which can be used intuitively and independently by patients but yet allows the accurate and reliable measurement of muscular activation. The sEMG sensor system introduced here, fulfills the SENIAM recommendations for sEMG recording and it will be shown that it is not sensitive to incorrect positioning. Therefore, it enables the autonomous application by patients while allowing a reliable detection of the muscular activation of different muscles during everyday tasks and therapeutic exercises.

## 2. Materials and Methods

### 2.1. sEMG Sensor System

#### 2.1.1. Overall Design of the Sensor System

The sEMG sensor system was designed to be a simple-to-use and size-adaptive device. Its properties should allow for the user to independently put on and take off the device even in the presence of reduced arm function. Since laypersons cannot ensure correct positioning of the sEMG electrodes, a multichannel approach was chosen. The multichannel approach assumes that if the electrodes are randomly positioned by the user, the likelihood that at least one sEMG channel is sufficiently correct positioned increases. Therefore, the sEMG sensor system consists of a total of 8 sEMG sensor modules, which are arranged in rotationally symmetrical ring form (Figure 1A). Each sEMG sensor module provides two bipolar sEMG leads allowing a total of 16 sEMG channels to be acquired with the device. As in the case of the Myo Armband or the 3DC Armband [21], the rotational symmetry of the sensor system leads to a reduction of the moment of inertia acting on each individual sEMG module. Displacements of the sensors around the bony axis of the limb, which normally occur during the execution of movements due to the inertia of the sensors and the suppleness of the soft tissue, are prevented by the symmetrical arrangement.

To ensure that the sEMG sensor system can be positioned and operated by any patient autonomously, the sEMG sensor modules are connected by extension springs (Figure 1A). When the springs are released, the armband is expanded and can be easily pulled over the limb in one step. Moreover, the modules are supported by two elastic bands that are parallel to each other and pass through both the extension springs and the housing of the modules. This ensures that the extension springs are held in a position perpendicular to the sEMG sensor modules. The length of the elastic bands can be shortened by turning the rotary knob (Figure 1A). This reduces the circumference of the sEMG sensor system and the eight sEMG sensor modules will be pressed steadily and securely onto the skin surface. The elasticity of the bands allows the sensor system to adapt to changes in the shape of the limb occurring during the performance of everyday activities or therapeutic exercises while maintaining the sEMG sensor modules’ position relative to the skin surface. Additionally, by shortening the bands the extension springs are tensioned. This causes a spring-load to act on the adjacent sensor modules, which provides additional support to stabilize the position of the modules.

A data cable and power supply runs through all sensor modules (Figure 1A). It is flexible and strain-relieved so that no additional forces and torques are applied to the sEMG sensor modules. The sensor module adjacent to the rotary knob houses the electronic circuitry necessary for digitalization and further processing of the signals as well as the option to accommodate a battery (Figure 2). sEMG signals are sampled with two 16-bit Analog–Digital Converters (ADS8332; Texas Instruments) and a sampling frequency of 2000 Hz each. The signals are subsequently further processed by a microcontroller (EFM32WG 380 F 256, Silicon Labs). The microcontroller runs the algorithms for identifying the relevant sEMG channels (Section 2.1.3) and implements the protocols for either wired or wireless transmission to an external storage device (Figure 2).

The sEMG sensor system can be effortlessly released by pressing the rotary knob. The spring force of the extension springs pushes the sEMG sensor modules apart and the elastic bands unroll from the rotary knob. This increases the circumference of the armband and it can be easily removed from the limb. Figure 1B shows the application of two sEMG sensor systems applied to the upper arm and to the forearm. The rotary knob and extension springs allow the sEMG sensor system to be operated completely with one hand. These properties enable the user to independently put on and take off the armband, even with impaired arm function.

#### 2.1.2. Design of the sEMG Sensor Modules

Each individual sEMG sensor module has a size of 15 mm × 25 mm × 50 mm (height × width × depth) and is equipped with seven dry cap-electrodes. The cap-electrodes are gold plated and have a hemispherical shape. The active surface of the electrodes has a diameter of 8 mm and protrudes above the case of the sEMG sensor module by 2 mm (Figure 3A). The shape and size of the cap electrodes ensure that the electrodes do not lose contact with the skin surface even under highly dynamic movement conditions.

Since a correct localization of the sEMG electrodes cannot be ensured when used autonomously by the patient, each sEMG sensor module provides two sEMG channels. These are obtained by three active electrodes arranged along the center axis of the sensor module (Figure 3A) building two bipolar leads (Figure 3B). In accordance to the SENIAM recommendations, the chosen inter-electrode distance is 20 mm [35]. Because of the ring-shaped geometry of the sEMG sensor system, the individual sensor modules are aligned with their central axis parallel to the bony axis of the limb (Figure 1B). As a result, one of the two sEMG channels of each module is more proximal located and the second more distally. This minimizes the error associated with the relative position along the longitudinal axis of the limb.

To reduce motion artifacts and noise, the sEMG signals are pre-processed directly at the electrodes. For this purpose, each sEMG sensor module contains its own circuit (Figure 3B). An impedance converter is used to compensate for the high impedance at the interface between skin and electrode. The signal then undergoes × 10 preamplification by means of a differential amplifier. Here the central electrode serves as an input for both differential amplifiers. This is followed by a high pass filtering with a first order Bessel filter and a cut-off frequency of 10 Hz. The high-pass filter eliminates motion artifacts caused by any relative movement of the electrodes to the skin surface in order to further amplify the best possible signal. The main amplifier has a gain factor of 100 and, together with a level adjustment, it adjusts the signal amplitude to the input voltage range of the analog-to-digital converter. To avoid aliasing, the sEMG signals are low-pass filtered with a cut-off frequency of 500 Hz (second order Bessel filter). The pre-processed sEMG leads are then passed to the A/D converter for digitization (Figure 2). Each sEMG channel is sampled at 2000 Hz with a 16-bit resolution (Figure 3B).

A special challenge is represented by the ground electrode, since all interferences that occur at the ground electrode are reflected in each sEMG channel. This is especially the case for motion artifacts. For this reason, each sEMG sensor module is equipped with four additional cap-electrodes, which are symmetrically positioned in a rectangular arrangement on the outside of the active electrodes array (Figure 3A). Like the active electrodes, the ground electrodes have an electrode distance of 20 mm to each other. The ground electrodes of all eight sEMG sensor modules are short circuited and form the common reference of the entire sensor system to which all electronic components are aligned. The reference is thus formed by a total of 32 cap-electrodes, which are arranged symmetrically around the limb. This results in a large equipotential surface, which stabilizes the common reference and reduces the influence of disturbances at the individual ground electrodes.

#### 2.1.3. Identification of the Relevant sEMG Channels

Rotational symmetry and the high number of sEMG channels increase the likelihood, that at least some sEMG channels are correctly positioned and aligned to the muscles for which muscular activation is to be determined. Since the sEMG sensor system provides 16 bipolar sEMG channels, it is necessary to determine those channels that best represent correct muscle localization. Identification is done by the microcontroller (Figure 2). Only the signals of those sEMG channels selected as relevant are then transferred to an external storage device.

The first step in identifying the relevant channels is to determine the number of muscles from which the muscular activation is to be recorded. Afterwards, an arbitrary calibration movement has to be performed by the subject. All 16 sEMG signals, derived during the calibration movement, are band pass filtered (Butterworth, ninth order, 20–500 Hz), full wave rectified and smoothed with a moving average 80 ms sliding window to obtain the envelope of the sEMG signal. The envelopes resulting from each of the 16 sEMG channels are divided into epochs of 80 ms duration. Afterwards, the root mean square (RMS) value of all envelope samples belonging to a single epoch is obtained (Figure 4).

The assumption that an appropriate electrode location relative to the excited muscle is characterized by a high and stable RMS value throughout the movement, suggests a hierarchic approach to identify the relevant channels. Therefore, for each epoch the 16 sEMG channels are ordered hierarchically starting with the highest RMS value. The sEMG channel that reached the highest RMS value in most epochs, was identified as the first relevant channel (Figure 4). In order to identify additional sEMG channels that reflect the activation of other muscles involved in the movement, the procedure was repeated step by step for the next hierarchical levels. If two sEMG channels identified as relevant originate from the same sEMG sensor module, the channel with a lower RMS value in most epochs is discarded. In this manner, the algorithm is less affected by outliers since it selects the channel that occurs most frequently in all epochs.

It is obvious that the described procedure is not suitable to assign a specific muscle to each sEMG channel. The detection of 16 sEMG channels was motivated by the fact that the sEMG sensor system should be applied independently by the subjects and that a greater number of derived sEMG channels, increases the likelihood that at least a few of them are correctly positioned relative to the muscles or muscle groups involved in the movement. Therefore, the approach presented here is rather intended to identify one, maximal three sEMG channels to which the activation of individual muscles can be reliably assigned. The number of channels to be considered as relevant ones depends on the number of muscles from which a sEMG signal is to be derived or can realistically be derived. When the relevant channels have been determined by means of the algorithm for identification of the relevant sEMG channels, the microcontroller reads only these channels at the A/D converter and transfers them to the external storage device according to the data transmission protocol. This reduction of the sEMG channels from 16 to one or three enables the realization of high sampling rates and wireless data transmission for further processing.

### 2.2. Demonstration of the Proof-of-Principle of the sEMG Sensor System

The proof-of-principle of the sEMG sensor system was demonstrated using the example of a voluntarily performed elbow extension/flexion movement. The measurement protocol comprised two parts. First, the subjects were asked to position the sEMG sensor system by themselves at the level of the biceps on their dominant arm. Thus, the localization of the sEMG channels around the upper arm at the level of the biceps was arbitrary. No skin preparation was done before the assessment. The position of the sEMG sensor system was photographically documented at the start of each measurement. After putting on the sEMG sensor system and waiting for 60 s, the sEMG signals were recorded with the arm at rest over a period of 20 s. Based on this static measurement, the noise level of the sEMG sensor system was determined separately for each channel. Afterwards the subjects were strapped to a pulley machine (5 kg load) so that the axis of the elbow coincided with the central axis of a deflection pulley. In this way, a constant torque was ensured over the entire range of motion [38]. Subjects were next asked to perform five repetitions of an elbow flexion/extension movement with self-selected movement velocity. sEMG signals of all 16 channels were recorded during movement performance. Afterwards, the subjects were asked to remove the sEMG sensor system. Thus, the procedure consisted of four steps: independent positioning the sEMG sensor system, measurement of the signal’s noise level at rest, recording of the sEMG signals during elbow flexion/extension movements and removal of the device. This four-step procedure was repeated five times for each subject. A 1-min pause was left between repetitions of the procedure to avoid fatigue.

From the 16 recorded sEMG channels, the first relevant sEMG channel was determined for each positioning of the device according to the algorithm described in Section 2.1.3. Since in the upper-arm the biceps muscle is mainly active during an elbow flexion/extension movement, it can be assumed that the first relevant sEMG channel represents the biceps EMG. An experienced user then verified whether the sEMG channel selected by the algorithm matched the one closest to the position recommended by SENIAM for biceps sEMG recordings [35].

The second part of the protocol involved the use of a commercial sEMG system (Noraxon USA, Inc. Scottsdale, AZ, USA; sampling frequency of 1500 Hz, 1 channel, gain of 1000) using self-adhesive Ag-AgCl bar shaped pre-gelled surface electrode pairs with an inter-electrode distance of 2 cm (Ambu^®^ Blue Sensor N, Ambu GmbH, Bad Nauheim, Germany). The electrodes were placed on the biceps by an experienced user following the recommendations of SENIAM. After preparation, the noise level at rest as well as the biceps sEMG during five trials with five repetitions of elbow flexion and extension movements each was measured.

In order to compare the sEMG sensor system with the system using pre-gelled electrodes, the biceps sEMG signals of both systems were band pass filtered (Butterworth, ninth order, 20–500 Hz), full wave rectified and smoothed with a moving average 80 ms sliding window to obtain the envelope of the sEMG signal. Mean plus 10-fold standard deviation of the sEMG envelope was chosen as a threshold to determine the phases of muscular activation in the sEMG signal. The sEMG envelope’s RMS was calculated during phases of muscular activation in order to quantify signal energy. In addition, the signal-to-noise-ratio (SNR) was determined by relating the RMS of the sEMG envelope during the phases of muscular activation to the RMS of the sEMG envelope resulting from the measurement at rest.

The measurements were performed by 10 healthy subjects (5 males, 5 females, age 41 ± 12.89 years old). Exclusion criteria included upper limb disorders and left-handed subjects. In the case of the sEMG sensor system this results in 50 (5 positions × 10 subjects) different positions of the device, combined with a 50-fold determination of the first relevant channel, 50 RMS values for the phases of biceps activation in the first relevant SEMG channel and a 50-fold determination of the SNR of the system. The study was approved by the Human Ethics Committee of the RWTH Aachen University and all subjects gave written informed consent prior to the study.

## 3. Results

All subjects included in the study were able to put on and take off the sEMG sensor system independently. The position at which they located the armband corresponded to the SENIAM recommendations for sensor location on the biceps (one-third of a line from the fossa cubit to the medial acromion). The combination of extension springs, elastic band and rotary knob resulted in parallel alignment of the sEMG sensor modules with the bony axis of the upper arm in all cases. During the execution of the elbow flexion/extension movements, movement artifacts in the sEMG signals occurred only rarely, even at high movement speeds. The contact of the electrodes to the skin surface was maintained even if the form of soft tissue in the upper arm changed during the contraction and relaxation of the biceps. This issue was particularly important in the case of athletic subjects. None of the subjects complained about discomfort from the device during the measurements. Figure 5 compares the 16 unprocessed sEMG signals of the sensor system over five flexion/extension cycles with a sEMG signal recorded under the same movement conditions from the commercial sEMG system using pre-gelled electrodes. In this example, the first relevant channel representing biceps activation was channel number C6 (red Figure 5A). No difference in signal quality could be seen between the first relevant channel of the sEMG sensor system and the reference measured with the commercial sEMG system (Figure 5B). In only three of the 50 trials, the identification of the first relevant channel by the algorithm did not match the expert’s assessment (for details see Appendix A, Table A1). In these three cases, however, there was an agreement between the expert’s evaluation and the second automatically determined relevant channel.

Compared to the system using pre-gelled electrodes (mean RMS for all subjects: 0.08 ± 0.03 mV), in phases of muscular activation the RMS values of the first relevant channel of the sEMG sensor system (mean RMS for all subjects: 0.13 ± 0.06 mV) were generally higher (Figure 6). This difference resulted in a significance level of *p* = 2.6 × 10^−7^ when using a one-sided, paired Student’s *t*-test. As was to be expected, the RMS values varied considerably between the different subjects. However, with the exception of subjects no. 2 and no. 8, all other subjects showed a high RMS value in the first relevant channel that was matched by a high RMS value when using the commercial system (pre-gelled electrodes) (Figure 6). When considering each subject separately, it was noted that intra-individual RMS values vary more when the sEMG sensor system is used compared to sEMG signals recorded with the commercial system (Figure 6). This is reflected in the standard deviation of the RMS values (STD) over the respective 5 trials of each individual subject. The mean STD of the intra-individual RMS values was 0.028 ± 0.03 mV when the sEMG sensor system was used and 0.008 ± 0.011 mV when the signal was recorded with the commercial sEMG system (Appendix A, Table A2). This difference is significant at a level of *p* = 0.04 when using a one-sided, paired Student’s *t*-test.

Figure 7 compares the SNR of the first relevant channel of the sEMG sensor system during phases of muscular activation with sEMG signals recorded using the commercial sEMG system (pre-gelled electrodes) when electrode placement complied with to SENIAM recommendations. In 4 of 10 subjects the SNR for the case of pre-gelled electrodes was above the SNR when using dry electrodes; in 5 subjects the SNR was comparable for both sEMG systems and in 1 subject the SNR of the sEMG sensor system was above that of the commercial system. The mean SNR is 28.52 ± 12.53 dB if the signal is derived with the sEMG sensor system and 35.2 ± 13.72 if derived with the commercial system (Appendix A, Table A3). When using a two-sided, paired Student T-test a group difference can be detected at a significant level of *p* = 0.003. When the sEMG sensor system using dry electrodes was used, the SNR obtained was between 71.15 dB and 4.08 dB, whereas it was between 67.93 dB and 10.87 dB when using the system based on pre-gelled electrodes. Similarly to the RMS, there are large inter-individual differences in SNR between the individual subjects, whereby, with two exceptions (subjects no. 3 and no. 4), a high individual SNR value in the first relevant channel of the sensor system is always accompanied by a high SNR value when detected with pre-gelled electrodes and vice versa (Figure 7). Intra-individually, the SNR values vary more when the SEMG signal is detected by the sEMG sensor system than when it is derived with the pre-gelled electrode-based system where SENIAM recommendations are applied. The mean STD of the SNR is 9.11 ± 5.02 dB in the first relevant channel of the sEMG sensor system. Compared to this, the mean STD of the SNR is significantly lower (*p* = 0.009) when using the commercial sEMG system (pre-gelled electrodes) (mean STD 3.50 ± 3.49 dB) (Appendix A, Table A3).

## 4. Discussion

The main objective of this paper was the demonstration of the proof-of-principle of a sEMG sensor system, which can be used intuitively and autonomously by the patients and still allow a reliable and accurate detection of muscular activity, which is comparable to recordings complying with the SENIAM recommendations. These characteristics as well as the fact that the design of the sEMG sensor system ensures that it can be applied easily with one hand, provide the sEMG sensor system with great advantages compared to other armband designs mentioned in the introduction and even textile applications. The introduced device has a rotationally symmetric design of eight sEMG sensor modules, which provide a total of 16 bipolar sEMG channels. The gold coated, dry, cap electrodes have a diameter of 8 mm and an inter-electrode distance of 20 mm. Thus, the minimum requirements of the SENIAM recommendations for the electrode arrangement are fulfilled by the device [35]. The choice of the electronic components in connection with an automatic selection of the relevant sEMG channels allows the realization of sampling frequencies of more than 2000 Hz at a resolution of 16 bit. Accordingly, the relevant sEMG signals are available with a sufficiently high amplitude and temporal resolution. In particular, the impedance converter before pre-amplification of the sEMG signals supports the use of relatively small (diameter 8 mm) dry electrodes, which are necessary to enable a patient-autonomous usage.

In a total of 50 trials on 10 healthy volunteers it was shown that the device can be put on and taken off independently by the subject, even if one arm was not used to support the donning and removal process. Thus, it was demonstrated that the device can also be used by persons with (partial) paralysis of one limb. This is an essential requirement for applications in the field of rehabilitation. However, the most important prerequisite for autonomous usage by the patient is to ensure sufficiently correct localization of the sEMG electrodes relative to the muscle from which the sEMG signal is to be derived. Incorrect localization of the electrodes affects the signal amplitude as well as the signal dynamics and carries the risk of crosstalk from adjacent muscles [26,27,28]. To overcome this problem, a multi-electrode approach was used. The hypothesis was that if a sufficient number of sEMG channels is available, even if the electrodes are randomly positioned by the user, at least one sEMG channel would be positioned with adequate accuracy. For this purpose, eight sEMG sensor modules were distributed rotationally and symmetrically around the limb and this arrangement was repeatedly shifted parallel to the bony axis of the limb. The overall arrangement of two rings of bipolar sEMG leads per module of the sEMG sensor system that derive in the direction of the muscle fibers was intended to prevent the sEMG sensor system from being positioned at the wrong level of the limb (proximal/distal positioning). Based on the RMS of the resulting 16 sEMG signals, an algorithm was implemented that automatically identifies, in a hierarchical order, the relevant sEMG channels that reflect the activation of those muscles primarily involved in the movement. Using the example of biceps activation during elbow extension/flexion movement, it was shown that in 47 out of 50 trials, the algorithm identified that sEMG channel as relevant for the biceps, which an experienced sEMG user would also identify as the one that comes closest to the SENIAM recommendations in its position. Since the three remaining sEMG channels incorrectly identified by the algorithm were located opposite to the correct sEMG sensor module, it can be assumed that in these cases the algorithm actually identified triceps muscle activity. It must be admitted, however, that the automatic identification of biceps activity in an elbow flexion/extension movement is a relatively simple task for the algorithm, since only a few muscles are involved in the movement. How well the algorithm works for more complex movements needs to be investigated in further studies. The relatively simple scenario used here, however, demonstrates the feasibility of the approach and enables an analysis of the accuracy and reliability with which sEMG signals can be detected when using this sEMG sensor system, which is autonomously utilized by the user. Furthermore, the algorithm for identification of the relevant channels is particularly applicable for wireless solutions since it ensures that the data is wirelessly transmitted at high sampling rates. However, this was not part of the objectives of the study.

As was to be expected, the RMS values as well as the SNR of both methods vary considerably among the individual subjects. This is most likely due to subject specific differences such as skin impedance, distance between muscle and electrode or percentage of the maximum voluntary contraction (MVC) required to lift the 5 kg weight. Furthermore, an influence of a subject specific predisposition to crosstalk cannot be excluded. All these subject specific influences have effects on the RMS of the sEMG signal and therefore on the SNR, too [39]. In addition, the sEMG signal depends on various biomechanical factors such as movement velocity or co-activation of synergetic or antagonistic muscles, which were not controlled for this study [36,38]. Therefore, the intra-individual evaluation provides more meaningful information about the quality of the sEMG signals arising from the two methods of measurement. It was noted that intra-individually both the RMS values and the SNR vary more when using the sEMG sensor system (dry electrodes) than when using a sEMG system based on pre-gelled electrodes. This is probably related to the fact that the pre-gelled electrodes were attached once over the biceps according to the SENIAM recommendations, and then the five trials were performed without changing the electrode position again. On the other hand, the sEMG sensor system was repositioned before each trial. The higher variation of the RMS values and the SNR, therefore, reflects the influence of the electrode positioning. It thus becomes clear that a sufficiently correct localization of the electrodes of the relevant channel of the sEMG sensor system could be achieved in more than the half of the subjects. To achieve future improvements in this respect, a higher number of sEMG sensor modules, each with a smaller width than 25 mm is required. Electrode arrays, in which many electrodes are arranged two dimensionally at small distances, could improve this situation. However, there is no electrode array that can be independently applied by a patient using just one hand. Electrode arrays are therefore not yet available for autonomous utilization by the patient.

The proof-of-principle is given using the example of biceps activation during an elbow flexion/extension movement. However, the presented principle can be scaled up and down without any problems. Thus, by just adding more sEMG sensor modules, extremities with a larger circumference and more muscles can be analyzed. At the same time, an extension in distal/proximal direction is possible by enlarging the modules in longitudinal direction. The addition of further sEMG sensor systems enables the simultaneous examination of different segments of an extremity (Figure 1B) as well as several extremities.

Although there is a significant group difference in the RMS values as well as in the SNR, the respective values are in a similar range. This means that the signal quality achieved with the sEMG sensor system is comparable to that of a commercial system using pre-gelled electrodes. According to Sinderby, the signal quality of the sEMG signal is considered poor if the SNR falls below a threshold of 15 dB [40]. In the case of the sEMG sensor system, sEMG signals with insufficient signal quality were recorded in 13 of 50 trials (Appendix A; Table A3). Five of the 13 trials showing insufficient signal quality came from subject no. 10, for whom all five trials with pre-gelled electrodes also had an SNR below 15 dB. This subject was not able to completely relax the muscles of the upper arm during the resting measurements, so that the noise level could not be determined sufficiently. The situation is comparable for subject no. 2 and subject no. 4, where three of the five trials each had an SNR below the threshold of 15 dB. In both subjects, the SNR using a commercial system with pre-gelled electrodes is also below average. Three of the trials which displayed poor signal quality, when using the sEMG sensor system, corresponded to the case in which the channels were incorrectly assigned by the algorithm. Although, in these three cases, the second relevant channel was considered when calculating the SNR, this approach does not seem to be comparable. It can even be assumed that the algorithm has not led to a correct identification of the first relevant channel due to the lower signal quality. Considering the RMS value of trials with a poor SNR, it was noted that in most cases, the RMS value of these specific trials is below the mean RMS value for all subjects (Appendix A; Table A2). This in turn leads to the possibility that precisely in the measurements with poor SNR no satisfactory electrode placement could be achieved.

Surprisingly, the RMS values of sEMG signals obtained by the sEMG sensor system were significantly higher than RMS values of sEMG signals derived with the commercial system using pre-gelled electrodes. There is no direct explanation for this, since dry electrodes are usually associated with a higher electrode impedance [17,18]. Motion artifacts, due to the higher mass of the sEMG sensor system, did not occur during the measurements and can therefore be excluded as the reason. Cross-talk due to an inaccurate positioning of the electrodes could play a role, but does not explain the fundamental difference. The elastic bands that fix the sEMG sensor modules to the skin surface could play an important role. To ensure that there is no relative movement between the electrodes and the skin surface, a high contact pressure is required. This, and the fact that the cap electrodes, which extend 2 mm beyond the housing of the sEMG sensor module press into the connective tissue results in the electrodes being closer to the source and less tissue in between. Both these factors lead to an increase in signal amplitude. The second possibility is that the pre-gelled electrodes needed a longer time to lower the skin impedance, via the gel, than that which was available during the measurements. This is also supported by the fact that the RMS increases with increasing trial number when using pre-gelled electrodes. However, all of these considerations regarding the higher RMS values when using the sEMG sensor system are speculative and require further investigation.

## 5. Conclusions

The sEMG sensor system introduced has demonstrated, in experimental validation, its proof-of-principle that it is able to reliably obtain high quality signals even when applied by unexperienced users without detailed knowledge about the exact positioning of the sEMG electrodes and with (partial) paralysis of one limb. Electrode size and inter-electrode distance in each sensor module correspond to the SENIAM recommendations. The use of extension springs and elastic bands leads to a stable contact between the skin surface and the electrode and, together with the rotational symmetry of the device, limits movement artifacts even in movements with high acceleration. The chosen multi-channel approach ensures that the electrodes of a few of the 16 sEMG leads are positioned relative to the muscle with sufficient accuracy. To detect the relevant sEMG channels, a simple identification algorithm was implemented on the sensor system’s microcontroller. After identification of the relevant channels, only their data are processed in order to achieve a high temporal resolution of up to 2000 Hz. Despite the progress made, there is still room for improvement. First and foremost, it would be desirable to increase the density of the sEMG leads in relation to the circumference of the sEMG sensor system’s circular form. This would allow a more precise positioning of the electrodes relative to the activated muscle and thus contribute to both an improvement in SNR as well as a reduction of the risk of cross-talk. For this purpose, the use of electrode arrays would be appropriate, but it must be ensured that they can be independently utilized by patients with disabilities. Secondly, the experimental evaluation of the sEMG sensor system should be extended to more complex movements involving more muscles. It is to be expected that, under such conditions, the algorithm for identifying the relevant sEMG channels will need to be refined, including for the identification of malfunctioning electrodes. The mechanical and electrical components of the sEMG sensor system allow for these modifications and it would even be possible to use two sEMG sensor systems to obtain information from several segments of the limb (Figure 1B). Despite all its limitations, the demonstration of the proof-of-principle of the introduced sEMG sensor system positions it as a wearable device that has the potential to monitor muscular activation in home and community settings.

## Figures and Tables

**Figure 1 sensors-20-07348-f001:**
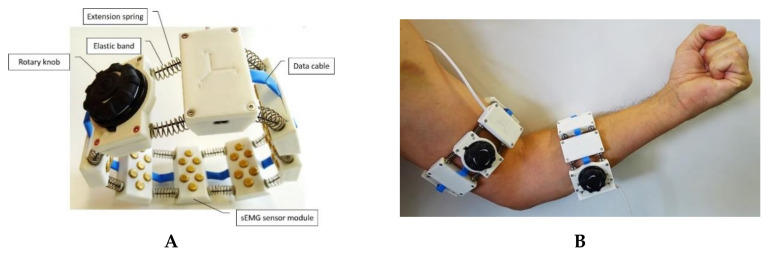
sEMG sensor system. (**A**) 8 independent sEMG sensor modules are arranged in a ring to form a rotation-symmetrical armband. The individual sEMG sensor modules are connected by elastic bands and extension springs. The size of the sEMG sensor system can be individually adjusted by the rotary knob. (**B**) An example showing the application of two sEMG sensor systems for monitoring the muscular activation of upper arm and forearm muscles.

**Figure 2 sensors-20-07348-f002:**
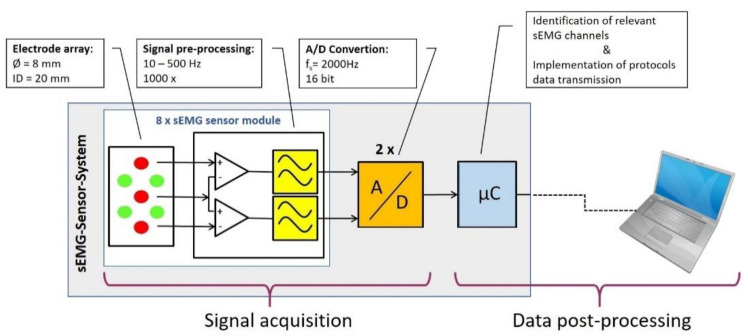
Block diagram of the sEMG sensor system. Integral part of the sensor system are the eight sEMG sensor modules which comprise 16 active sEMG electrodes with integrated preprocessing of the signals, two A/D converters and a microcontroller which is used to identify the relevant sEMG signals and implement the data transmission protocols.

**Figure 3 sensors-20-07348-f003:**
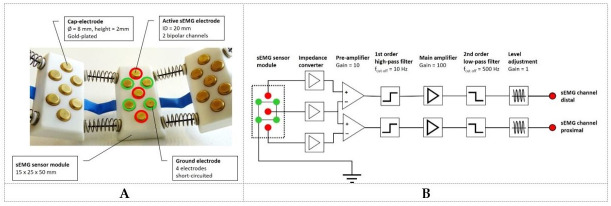
sEMG sensor module. (**A**) Each sEMG sensor module consist of seven gold-plated cap-electrodes. Four of them are short circuited to a ground electrode (green). The three active electrodes (red) are arranged along the center axis of the sensor module. They allow the recording of two bipolar sEMG channels, one distal and one proximal parallel to the bony axis of the limb. (**B**) Each individual sEMG sensor module incorporates a signal pre-processing unit that provides two sEMG channels for subsequent analog-to-digital conversion.

**Figure 4 sensors-20-07348-f004:**
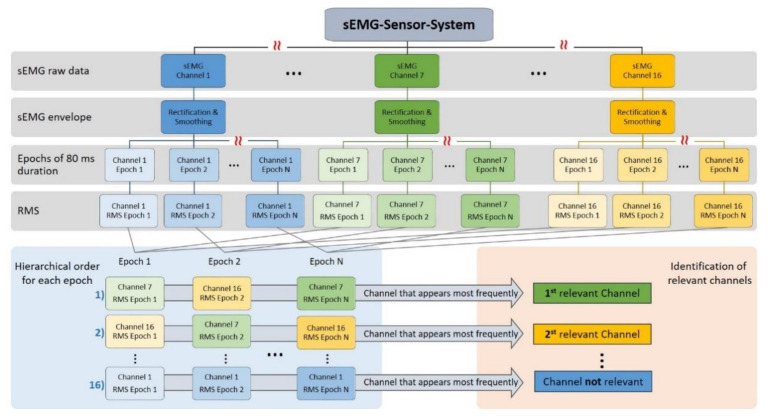
Block diagram of the algorithm for identifying the sEMG channels that best reflect muscle activation: the relevant sEMG channels.

**Figure 5 sensors-20-07348-f005:**
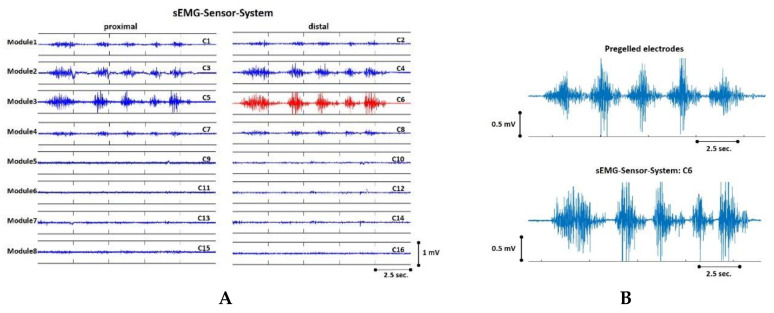
(**A**) Representation of the 16 sEMG signals of the sensor system. The sEMG signals were based on five repetitions of an elbow flexor/extension movement. Channel C6 of the sensor system was identified by the algorithm described in Section 2.1.3 as the first relevant channel, which in this case prevents biceps activation. (**B**) Comparison of the first relevant channel C6 with a sEMG signal of the same subject recorded with a commercial sEMG system using pre-gelled electrodes.

**Figure 6 sensors-20-07348-f006:**
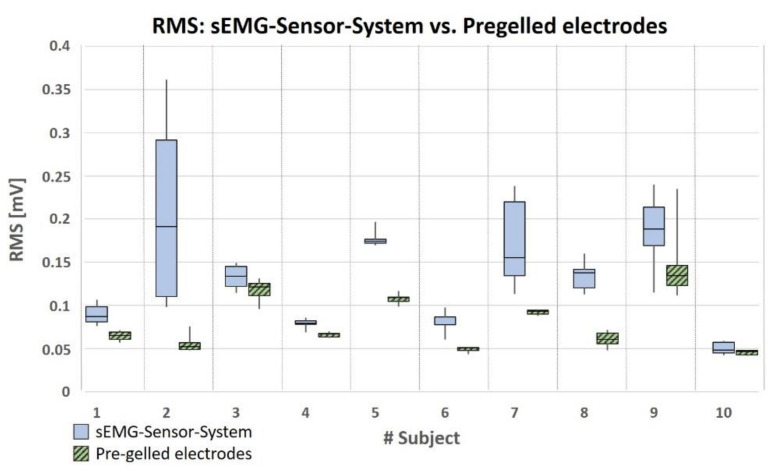
RMS values in the phases of muscular activation resulting from the sEMG signal of the first relevant channel of the sEMG sensor system and a sEMG signal derived with the commercial sEMG system (pre-gelled electrodes), all recorded according to the SENIAM recommendations. The values are based on five independent trials each of five repetitive flexion/extension movements of the elbow.

**Figure 7 sensors-20-07348-f007:**
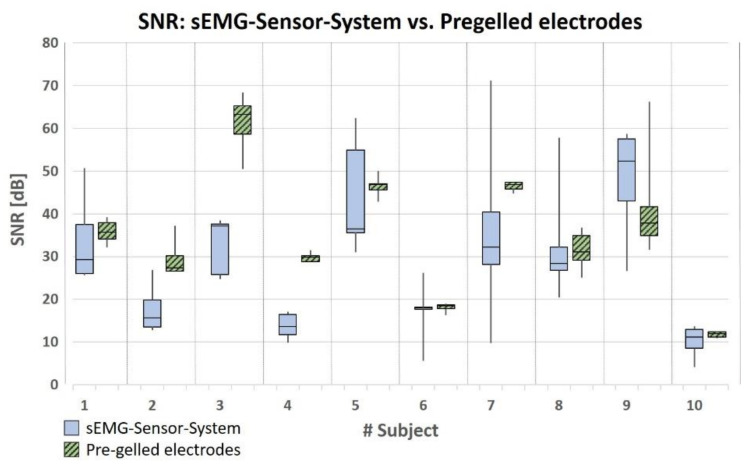
Signal-to-noise ratio (SNR) during the phases of muscular activity of the biceps muscle for the sEMG signal of the first relevant channel of the sEMG sensor system (dry electrodes) and a sEMG recording with the commercial sEMG system (pre-gelled electrodes). Electrode placement and signal processing complied with the SENIAM recommendations.

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
