# Peer review of "Introduction of a sEMG Sensor System for Autonomous Use by Inexperienced Users"

_sensors, 2020, doi:10.3390/s20247348_

Round 1

Reviewer 1 Report

This paper introduces sEMG-Sensor-System, which enables patients to apply autonomously and at the same time allows reliable detection of activation of different muscles in daily tasks and therapeutic exercises. The research content of this paper stands on the perspective of human needs and provides great help for patients with movement disorders. The topic is novel and the work load is sufficient. Some Suggestions are as follows:

  1. In 2.1.2, what is the deficiency if you have SI-interpolation forreconstruction? It is better to compare the advantages and disadvantages with the second order Bessel filter method in detail.
  2. In 1.3, how do you reduce the number of 16 sEMG channels to 1 or 3?Please describe the specific method.
  3. In 2,when performing elbow flexion/extension movement on subjects, are there any specific requirements for the subjects’ age and health status?
  4. In 2.2,is 10 subjects too few? Would the results be more convincing by increasing the number of healthy subjects and widening the age range of subjects?

Author Response

Dear Reviewer 1:

Thank you.

Elisa Romero Avila

Reviewer 2 Report

The authors reported their attempt at building and validating an easy-to-use and high-quality sEMG array with the idea of making it available for independent use outside the lab by inexperienced users. Even though not completely original (but the authors fairly commented on this), the idea is topical and of great potential for the scientific community working on real-life conditions. The theoretical focus on signal’s quality is as admirable as the results are. Such a device (or potentially resulting family of devices) embodies, in my opinion, the future of research and this paper is an undoubtedly useful contribution to the field. My few concerns, detailed below, are more on the fairness of the language used than on the experimental design. The authors correctly reported the limitations of the hardware in its present form and discussed them. However, thoughts on some additional concerns are missing. In summary, I propose to soften certain statements in order to make even the inexperienced reader aware of the limitations that come with such a design.

MAJOR COMMENTS

  1. The authors state several times that one of the major advantages of this device inspired to the Myo Armband is that (from the abstract) “It offers convenient features for the user, such as […] no prior knowledge about positioning of the sEMG electrodes”. While I reckon the undisputed expertise of the authors in the field of electromyography, in my opinion this sentence (and this approach at large) are misleading for several reasons. First, in order to follow the very same SENIAM recommendations that the authors carefully report of having followed for electrode characteristics, the authors acknowledge in the introduction that electrode positioning strongly influences the quality and nature of the signal. This means that the user does need to be instructed beforehand, since the device can be easily misplaced over irrelevant portions of the limb due to its design. A requirement that amplifies when working with patients or older adults (something I am aware of because of the experience in our lab with the Myo Armband) and that might undermine the requirements for the validation of positioning (lines 270-274). Second, from the point of view of the researcher that eventually will make use of the data, recording from eight equidistant, rotationally symmetric, electrodes in a confined and constrained region of the limb might limit the usability of the data. Modern approaches in motor control that look into muscle activity patterns such as, but not limited to, that of muscle synergies1 or uncontrolled manifold2, do require readings from several different groups of flexors/extensors, agonist/antagonist, possibly belonging to body segments larger than the upper limb. Also robotics applications, already referenced to in the text, would have similar requirements. The device in its current configuration poses a great limit to the applicability to those frameworks and this is not fully acknowledged in the manuscript. Third, the algorithm for the identification of the relevant sEMG channels would be of less interest if the sensors could be integrated in a wearable piece of clothing, with predefined landmarks. This would indeed pose production issues (e.g., finding the right materials, producing different sizes, etc.) but would dramatically reduce the positioning uncertainties as well. Also, it would avoid the need for the “arbitrary calibration movement” that, again, is something the user should be instructed about. A lengthy comment only to say that, for all those reasons and possibly others that the authors certainly already addressed during the design process, the writing on the easiness of positioning could be “smoothened” a little. This would make for a fairer presentation of this yet valuable work.

MINOR COMMENTS

  1. The manuscript gives little detail on the modalities for data transfer. While in lab conditions there can always be a researcher ready to activate the wireless transfer protocols, it would be useful to report the feasibility of such operations in home and community settings, especially given the focus of the paper on the out-of-the-lab applicability of the device.
  2. Line 247-255: this means that, if one electrode fails and sends full-scale signals to the algorithm, it would be identified as the first relevant channel? If so, wouldn’t some sort of normalisation and/or minimum subtraction (or similar) be needed to deal with defective sensors?
  3. Line 291-293: related to my major comment. The band could rotate/slide during daily activities and the fact that “an experienced user” was needed to verify “whether the sEMG channel selected by the algorithm matched the one closest to the position recommended by SENIAM” is a downside that should be discussed.
  4. Related to my comment above: could the algorithm be run on selected chunks of the 80-ms-epoch data set so that rotations of the device could be taken into account and adjusted for? This would be extremely relevant for long-term recordings in real-world settings.
  5. Line 304: why moving average and not any kind of zero-lag filter?

References

  1. Bizzi, E. & Cheung, V. C.-K. The neural origin of muscle synergies. Front. Comput. Neurosci. 7, 51 (2013).
  2. Tresch, M. C. & Jarc, A. The case for and against muscle synergies. Curr. Opin. Neurobiol. 19, 601–7 (2009).

Author Response

Dear Reviewer 2:

Thank you.

Elisa Romero Avila

Round 2

Reviewer 1 Report

The authors have a better modification for last review. It should be accepted.

Reviewer 2 Report

The authors addressed all my comments in the reply and implemented modifications to the manuscript following almost all my recommendations, with convincing answers (and motivations for disagreeing, where relevant). I would like to thank them for their efforts.